# Epidemiology of *Cryptosporidium* Infection in Romania: A Review

**DOI:** 10.3390/microorganisms11071793

**Published:** 2023-07-12

**Authors:** Gheorghe Dărăbuș, Maria Alina Lupu, Narcisa Mederle, Rodica Georgiana Dărăbuș, Kalman Imre, Ovidiu Mederle, Mirela Imre, Ana Alexandra Paduraru, Sorin Morariu, Tudor Rares Olariu

**Affiliations:** 1Discipline of Parasitology, Faculty of Veterinary Medicine, University of Life Sciences King Michael I, 300645 Timisoara, Romania; gheorghe.darabus@fmvt.ro (G.D.); kalmanimre@usab-tm.ro (K.I.); mirela.imre@gmail.com (M.I.); sorin.morariu@fmvt.ro (S.M.); 2Discipline of Parasitology, Department of Infectious Disease, Victor Babes University of Medicine and Pharmacy, 300645 Timisoara, Romania; drbsgeorgiana@yahoo.com (R.G.D.); mederle.ovidiu@umft.ro (O.M.); paduraruale@gmail.com (A.A.P.); rolariu@umft.ro (T.R.O.); 3Center for Diagnosis and Study of Parasitic Diseases, Department of Infectious Disease, Victor Babes University of Medicine and Pharmacy, 300041 Timisoara, Romania

**Keywords:** *Cryptosporidium* spp., Romania, humans, animals, descriptive and molecular epidemiology

## Abstract

Since 1983, when the first report of a human *Cryptosporidium* spp. infection was published in Romania, and until now, many studies on cryptosporidiosis have been published in our country, but most of them are in the Romanian language and in national journals less accessible to international scientific databases. Although the infection was first recognized as a problem in children or immunocompromised people or more of a problem in low-income or underdeveloped global countries, we have shown in this review that it can also occur in people with normal immunological function and that the epidemiology of our country can provide a theoretical basis for the formulation of a *Cryptosporidium* spp. prevention strategy. In addition, 9.1% of healthy children and 73% of immunocompromised children were observed to have *Cryptosporidium* spp. infections. Higher rates have also been reported in immunocompromised adults (1.8–50%). Analyzing the prevalence of *Cryptosporidium* spp. infection in animals, we found values of 28.52% in cattle, 18% in buffalo calves, between 27.8 and 60.4% in pigs, 52.7% in dogs, and 29.4% in cats. Furthermore, in Romania, the burden of cryptosporidiosis, including acute infections and long-term sequelae, is currently unknown.

## 1. Introduction

*Cryptosporidium* was first recognized in the gastric glands of the laboratory mouse by Tyzzer in 1907, and in a preliminary description dated in 1910, it was given the name *Cryptosporidium muris* [1]. Later in 1912, Tyzzer described a second species called *Cryptosporidium parvum*, also identified in mice, that differs from the previous one not only by the fact that it infects the small intestine rather than the stomach but also in the size of the oocysts, which are smaller [2]. In the following years, *Cryptosporidium* was identified in various tissues and organs in mammals, birds, fish, and reptiles.

*Cryptosporidium* was recognized as an important enteric pathogen in humans for the first time in 1976 [3]. Since then, nearly 20 *Cryptosporidium* species that can infect humans have been identified, with *Cryptosporidium hominis* and *C. parvum* being the most frequently detected [4]. Even though *C. parvum* is the most common in some countries, such as Lebanon, Israel, Egypt, and Tunisia, *C. hominis* was predominant [5]. Over the last 50 years, many studies have caused confusion between this genus and other genera of Apicomplexa, and several new species of cryptosporidia were erroneously named or classified [6]. Recently, the molecular characterization of cryptosporidia has helped to clarify the confusion encountered in the taxonomy of these protozoa.

In Romania, the first documented reports of *Cryptosporidium* spp. infection were made by Dan et al. [7] in calves (prevalence 33–100%) and Olteanu (1983) in humans, cattle, goats, sheep, pigs, deer, rabbits, mice, and rats. In Timiş County, Western Romania, between 1988 and 1995, coprological samples were examined from ten species of mammals (cattle, sheep, pigs, horses, rabbits, dogs, cats, mice, rats, and guinea pigs), and the natural infection was observed only in cattle and pigs [8]. Prevalences in different animal species, genotypes, and subtypes of *Cryptosporidium* were later reported by Dărăbuş et al. [9]. The prevalence of *Cryptosporidium* in soil is directly influenced by several factors: continent, air pressure, temperature, and detection methods. These factors must be individualized from country to country [10]. The aim of the present paper was to review the epidemiological data regarding the *Cryptosporidium* spp. studies carried out in Romania, highlighting the currently available information on its epidemiology, genetic diversity, and distribution. Although the PUBMED Database indicated only 18 referenced papers on cryptosporidiosis in humans and animals in Romania, we found numerous studies published in Romanian journals. We have also found three doctoral theses on cryptosporidiosis in animals and one thesis on human cryptosporidiosis to incorporate all full-length available reports regarding this zoonosis. Abstracts presented at conferences or symposia and studies providing incomplete data were not included in this review.

## 2. Results

### 2.1. Humans

There is limited information in the international scientific literature on *Cryptosporidium* prevalence in the Romanian population, as it is mainly published in national journals. In Romania, *Cryptosporidium* spp. screening is not a routine component of coproparasitological standard examinations in clinical laboratories; therefore, most studies are surveys based on convenient samples collected from hospitalized children or adults. Most infections were diagnosed by direct microscopic stool examination [11,12] and/or modified Ziehl–Neelsen staining [11,13,14,15,16,17]. Other diagnostic methods were enzyme-linked immunosorbent assays (BIO K 070, BIO K I55—Bio-X Diagnostics, Jemelle, Belgium) [11,12], immunochromatography qualitative assays (RIDA Quick Cryptosporidium/Giardia Combi) [18], and immunofluorescence assays (Meridian) (Table 1). It is important to prevent disease transmission by reducing and eliminating oocysts. It is imperative to isolate animals and disinfect surfaces [19]. With the advancement of technology, new kits have appeared for the diagnosis of pathology. New techniques are based on fluorescence microscopy and immunological/molecular techniques [20].

### 2.2. Cryptosporidiosis in Children

The first report of human *Cryptosporidium* infection in Romania was published in 1983, showing a prevalence of 9.1% in healthy children and 12.3% in children with diarrhea. However, the prevalence of the infection in children varied widely, from 1.87% to 73%, depending on the area of residence, age, immune status, and method used for diagnosis (Table 2). In stool samples, *Cryptosporidium* spp. was identified in 3.2% of hospitalized children with acute diarrheal disease [26] and in 3.03% of children with physical and psychic problems [27]. In hospitalized children from Timis County, the infection prevalence was 4.26% in 2010 [28] and 7.54% in 2015 [12]. *Cryptosporidium* IgA or IgG antibodies were identified in 72–73% of children hospitalized at the Dystrophic Center and Colentina Hospital of Bucharest. In a recent published survey, Popa and colleagues suggested a declining trend in the seroprevalence of IgG antibodies [24].

No case of infection was observed in patients hospitalized at the Emergency Clinical County Hospital in Cluj-Napoca [18]. The highest prevalence (16.66%) of Cryptosporidium oocysts in feces was observed in hospitalized patients from Timiș and Arad counties [15]. Unfortunately, the frequency of cryptosporidiosis is increasing. It is a leading cause of diarrhea and malnutrition in children, regardless of their country of origin [32].

The infection rates of Cryptosporidium spp. in immunocompromised individuals are highly variable (1.8–50%) among different reports. Cojocaru and Cojocaru [30] identified a 1.8% prevalence of Cryptosporidium spp. in deceased HIV-infected children, and Brannan et al. (1996) demonstrated the presence of Cryptosporidium oocysts in 18% of the stool samples collected from hospitalized HIV-infected children. Moreover, Brannan et al. (1996) observed that the presence of Cryptosporidium IgA antibodies was correlated with HIV seropositivity and malnutrition severity, while IgG antibodies were linked only with severe malnutrition. In Timis County, the prevalence of Cryptosporidium infection among positive hospitalized children with HIV varied from 30% [15] to 50% [12]. Of 8 positive hospitalized adults with HIV from Arad County, 4 were found to be Cryptosporidium infected [15]. A study conducted in South Africa observed a high prevalence of Cryptosporidium spp. in immunocompromised patients. The spread of the pathology is significant in that region. Patients present with diarrhea and nutritional deficiencies [33].

### 2.3. Animals

Starting in 1982, when Cotofan used the histopathological examination to identify lesions in *Cryptosporidium* infection, several diagnostic methods were used in Romania to identify cryptosporidiosis, especially for research purposes [34].

Thus, coprological methods, with or without staining, were useful to determine the presence of *Cryptosporidium* spp.: HEA, PAS, Giemsa, toluidine blue stains, Direct microscopy, and Ziehl–Neelsen staining methods modified by Henriksen [22,35]. Other methods, including enzyme-linked immunosorbent assay [15], electron microscopy [8], and PCR [17,25] (Table 3), were utilized in several scientific studies. Of note, the coproparasitological examination and other specific methods have not become routine diagnostic tools in veterinary medicine.

### 2.4. Cattle and Buffalo

During 1983, cryptosporidiosis was found in calves aged 3–24 days with diarrhea, with a prevalence ranging from 33% to 100% [7]. In the same year, *Cryptosporidium* spp. was reported by Olteanu in 18 species of mammals, including calves with diarrhea (67.3%). The same author found a much lower prevalence in Bos bubalus (16.67%). In the following decade, investigations carried out in the western part of Romania reported a prevalence of 38.4% in cattle aged between one day and two years [8]. The extent and intensity of parasitism were higher in animals 8–14 days old [25,38]. Imre et al., in an epidemiological study carried out on 25 farms in western Romania between 2005 and 2009, reported a 41.6% prevalence in calves under the age of six months [22,37]. In another study of calves from central and north-west Romania, a prevalence of 25% was reported [39].

In a serological study carried out in 100 calves, 4 days to 5 months old, from Arad County, Imre et al. [40] found a prevalence of 63% (single infection or in association with other enteropathogens) (Bio-X Easy-Digest Antigenic ELISA Kit). The other enteropathogens were coronavirus and rotavirus. The same authors, analyzing by ELISA the fecal samples collected from calves in the first 4 weeks of life on eight industrial farms, found a prevalence of 64.36%. Other enteropathogens were also identified, such as coronaviruses and rotaviruses [41].

Bejan et al. (2008) identified *Cryptosporidium* in calves with severe digestive disorders. The general prevalence was 28.52%, with a higher prevalence in the age group of 1–3 weeks. Regarding the season, the prevalence was higher in late winter and early spring. A study, performed by copro-ELISA test in western, central, and northwestern Romania in calves aged between 1 and 30 days, with or without diarrhea, identified polyparasitism with *Cryptosporidium* (41.4%), rotaviruses (16.2%), coronaviruses (10.3%), and enteropathogenic *Escherichia coli* F5 (K99) (1.08%) [42,43] (Table 4).

Recently, in Romania, cryptosporidiosis was identified in buffalo calves with a prevalence of 18% [45].

### 2.5. Sheep and Goats

In southern Romania, in young sheep with diarrhea, cryptosporidiosis was diagnosed in 63.71% of animals out of 383 examined. The same author reported a prevalence of 89.34% in Capra hircus with diarrhea and 4.76% in those without diarrhea. In goats at pasture in the central regions of the country, the prevalence was 9% in spring, 20% in autumn, and 19% in summer. In youth, the proportion of animals parasitized with *Cryptosporidium* spp. reached 60% in autumn [46]. An epidemiological study carried out by ELISA in lambs in the first three weeks of life in western and northwestern Romania showed 9.1% *Cryptosporidium* infection [15]. Similar prevalences were found in newborn lambs by Imre et al. [47]. The study was carried out by processing through the Ziehl–Neelsen staining technique a total of 175 diarrheal fecal samples from lambs younger than 21 days. The molecular characterization of the isolates revealed the overwhelming dominance of *C. parvum*, followed by *C. ubiqitum* and *C. xiaoi*. (Table 5) In a study conducted in 41 countries, it was observed that sheep under 3 months of age have the highest prevalence [48].

### 2.6. Pigs

In 1983, Olteanu reported 60.4% of *Cryptosporidium* infections in Sus scrofa domestica in farms where digestive disorders had been reported in piglets. However, the first case of cryptosporidiosis in pigs was identified in western Romania in 1994 by Dărăbuş (but reported in 1996) after examining the scraping of the ileal mucosa in a pig that died of salt poisoning.

In the industrial-type farms from the west of Romania, Imre et al. [50], using the enzyme-linked immunosorbent assay ELISA, found a prevalence of 33%. Depending on the age, the infection ranged from 5.26% to 62.5%, with the most affected age group being 29–47 days. Recently, Băieș et al. [51] found a prevalence of 18.12% in weaning pigs and 9.38% in sows but no infection in fatteners in two free-range farms in the northwest of Romania (Table 5).

### 2.7. Dogs and Cats

Studies of canine cryptosporidiosis in the central area of Romania were carried out in the period 2008–2009 by Titilincu et al. [52]. After processing 374 fecal samples using the ELISA enzyme immunoassay technique, a prevalence of 52.7% was found. Risk factors were young age, rural areas, living conditions, and other parasitic infections. There was a higher prevalence of shepherd dogs, police dogs, and other dogs in rural areas. No case of *Cryptosporidium* infection was found by Imre (2010) in dogs in the west of the country using ELISA and Ziehl–Neelsen modified by Henricksen (Table 6).

Following an epidemiological investigation in cats in central Romania, Mircean et al. [54,55] found a prevalence of 29.4% by the Henricksen-modified Ziehl–Neelsen staining method combined with the ELISA enzyme immunoassay.

### 2.8. Other Mammal Species

Cryptosporidiosis was identified in other mammal species: *Capreolus capreolus* (31.58%), *Cervus elaphus* (16.67%), *Ovis musimon* (36.36%), *Sus scrofa* (31.25%), *Lepus europeus* (59.65%), *Lepus timidus* (12.5%), *Mus musculus* (22.03%), *Oryctolagus cuniculus* (41.38%), *Myocastor coypus* (14.29%), *Cavia porcelus* (22.22%), *Rattus norvegicus* (30.51%), *Rattus rattus* (26.09). Animals infected with *Cryptosporidium* spp. had digestive problems (diarrhea) and were mainly in the youth age category.

### 2.9. In Birds

In a study conducted between 1980 and 1982, Olteanu reported *Cryptosporidia* infection in eight species of birds: *Columba livia*, *Corvus frugilegus*, *Gallus domesticus*, *Meleagris gallopavo*, *Numida meleagris*, *Passer domesticus*, *Paser montanus*, and *Phasianus colchicus*. Dan et al. [53] found that 15.4% of chickens aged 25–30 days and 10.4–21.2% of those aged 30–40 days were infected with Cryptosporidium. In turkeys, the prevalence varied between 25.7% in those aged 28 days and 31.6% in the age group of 30–40 days.

In the period 1988–1995, in a study carried out in Timiş County on 893 chickens, 27 turkeys, and 23 partridges, cryptosporidiosis was found only in chicken broilers, and prevalence varied between 22.5% and 25.2%. Increased mortality was reported in broiler farms when *Cryptoporidium* infection was associated with the infectious bursitis virus. This may signify a synergism between the two pathogens [8,56].

### 2.10. Cryptosporidium Species Identified in Romania

About 44 valid species and over 120 genotypes are currently recognized worldwide, mostly based on molecular studies [57,58]. In Europe, eight species (*C. parvum*, *C. andersoni*, *C. bovis*, *C. ryanae*, *C. felis*, *C. hominis*, *C. meleagridis*, and *C. suis*) have been identified in cattle in 2011 and three genotypes (*Cryptosporidium deer-like genotype*, *C. suis-like* genotype, and *Cryptosporidium pig genotype*) [44]. In addition, by analyzing the DNA sequences of *C. parvum*, three family subtypes (IIa, IId, and III) were identified from the 11 alleles (IIa-IIk). Another study [59], carried out in the Czech Republic, revealed parasitism with *C. suis*, *C. scrofarum*, *C. parvum*, *C. muris in pigs*, *C. muris*, *horse genotype*, and *C. parvum*, *C. tyzzeri* in horses, *C. felis* in cats, *C. avium* in red-crowned parakeets, and *C. testudinis*, and *C. ducismarci*, tortoise genotype III in tortoises.

In Romania, a study was published in 2007 [60] that showed the morphometric identification of oocysts isolated from calves aged from 4 days to 5 months. Based on the shape, measurements, and shape index [61,62], it was observed that the identified species were *Cryptosporidium parvum* and *C. andersoni.* In 2009, in the western part of the country, 12 isolates of *C. parvum* were subtyped by amplifying the GP60 gene, identifying a single-family subtype Iia [49].

A preliminary study of unweaned lambs led to the identification of *C. parvum* species by PCR-RFLP, suggesting a potential risk of zoonotic transmission [44]. In unweaned calves from nine farms located in the Banat region, *C. parvum* species were identified by nested PCR. Two subtypes were identified by GP60 gene sequence analysis: IIaA15G2R1 and IIaA16G1R1 (Imre et al., 2011). In the same area, 29 isolates from calves were molecularly analyzed for two loci (SSU Rrna and 60-Kd glycoprotein genes). *C. parvum* was identified with five subtypes: IIdA27G1 (*n* = 8), IIdA25G1 (*n* = 5), IIdA22G1 (*n* = 2), IIdA21G1a (*n* = 1), and IIaA16G1R1 (*n* = 1. In the same study, only one subtype was found in pigs: IIdA26G1 [17].

In birds, although PCR tests were not used to identify cryptosporidial species, based on location, morphological, and biological characters, the species *C. meleagridis* was found in broiler chickens [8].

## 3. Discussion

As we highlighted in this review, the presence of numerous transmission routes makes the epidemiology of cryptosporidiosis complex; moreover, other studies show that the investigation of both sporadic cases and outbreaks has contributed to a better understanding of risk factors and sources of infection [63,64].

In a recent meta-analysis conducted by Dong et al. [65], which included 221 studies published between 1 January 1960, and 1 January 2018, it was found that the prevalence of *Cryptosporidium* spp. infection in the global population was 7.6% (95% CI: 6.9–8.5). In Mexico, it was the country with the highest estimated prevalence of *Cryptosporidium* spp. infection (69.6%), following Bangladesh (42.5%), Nigeria (34.0%), and the Republic of Korea (8.3%) among general residents, patients, schoolchildren, and the healthy population, respectively. Dong et al. also observed that the estimated prevalence was high in people from low-income countries, in people younger than 5 years, and in non-urban residents, as well as in those with gastrointestinal symptoms. However, due to the strict exclusion criteria (restriction to the English language, no information on the study period, location, method of diagnosis, sample size, number of infected people, or even outbreak, laboratory report, or immunocompromised population), no study conducted in Romania was included in this meta-analysis, even if the PUBMED Database indicated 18 referenced papers on cryptosporidiosis in humans and animals in our country. So, we believed that the results of this study would yield important data about *Cryptosporidium* spp. infection and the investigation of sources of infection.

In Romania, the first report of human *Cryptosporidium* spp. infection was published in 1983 [29], showing a prevalence of 9.1% in healthy children and 12.3% in children with diarrhea; however, the higher prevalence of the infection in children was 73% (Table 2).

To elucidate the possible epidemiology of cryptosporidiosis in young children, an open cohort from a semi-urban area was followed for 2 years. Possible risk factors were tracked and recorded every month. This case-control study included 125 children with *Cryptosporidium* spp. diarrhea and an equal number of matched controls. The following sources of infection were identified: the presence of pigs (odds ratio (OR) = 2.5, 95% confidence interval (CI) 1.4–4.7) and dogs (OR = 2.1, 95% CI 1.0–4.2) in the household, and the child’s sex (OR for boys = 1.9, 95% CI 1.0–3.4) [66]. When they analyzed the prevalence of all cases of diarrhea in children from that area (more than 2000 children), they noted that 6% had symptoms due to *Cryptosporidium* spp. In a previous study conducted by us and analyzed in this review, with a similar number of patients (*n* = 212, children from the urban and rural areas of Timis County with primary symptoms), the fecal samples taken showed a prevalence of cryptosporidiosis of 7.54% (16/212). Children had contact with animals (cats, dogs, lambs, calves), were living collectively (kindergarten and school), and the source of drinking water was wells in rural areas and springs/tap water in urban areas [12].

The infection rates of *Cryptosporidium* spp. in immunocompromised individuals from Romania are highly variable (1.8–50%) among different reports. In the world, the results of studies investigating the prevalence of cryptosporidiosis in HIV-positive patients with diarrhea estimated a prevalence that differed quite a lot from each other, with variations ranging from 0 to 100% with a median of 32% [67].

Analyzing in this review the infection in animals in Romania, it was observed that in cattle with or without diarrhea, *Cryptosporidium* spp. infection was 28.52% [42] and in buffalo calves, a prevalence of 18% [45]. The global prevalence of infection in cattle varies greatly between countries and studies. For example, in pre-weaned calves in the UK, reported prevalence rates range from 28.0 to 80.0% [68]. Studies from other parts of the world have reported a prevalence of infection in pre-weaned calves ranging from 3.4 to 96.6% [69].

In a study conducted by Krumkamp et al., where fecal samples were collected from each child (≤5 years old, *n* = 197), sheep (*n* = 334), and cattle (*n* = 862), with or without symptoms of infection, 11 samples (5.6%) from children, 30 (3.5%) from cattle, and 42 (12.6%) from sheep were tested positive for *Cryptosporidium* spp. In Romania, the prevalence in young sheep was between 9.1% [15] and 63.71% [29]. In a recent meta-analysis of a total of 126 datasets included for the final quantitative analysis from 41 countries, it was shown that globally, sheep <3 months of age had a significantly higher prevalence of *Cryptosporidium* spp. infection (27.8%; 3284/11 938) than those aged 3–12 and >12 months. At the same time, the prevalence of *Cryptosporidium* in sheep with diarrhea was 35.4%, significantly higher than in sheep without diarrhea (11.3%) [48].

According to Chen et al., the highest prevalence of *Cryptosporidium* spp. infection in pigs was 40.8% (478/1271) in Africa. Post-weaned pigs had a significantly higher prevalence (25.8%; 2739/11 824) than pre-weaned pigs, fattening pigs, and adult pigs. The prevalence of infection was lower in pigs that had diarrhea (8.0%; 348/4874) compared to those without diarrhea (12.2%; 371/3501). In Romania, depending on the pigs age, the cryptosporidiosis ranged from 27.8–60.4% (Table 5).

In our country, studies of cryptosporidiosis found a prevalence of 52.7% in dogs and 29.4% in cats. The pooled global prevalence of *Cryptosporidium* spp. in cats was 6%, being highest in Africa at 14% (0–91%) and lowest in South and Central America at 4% (3–7%) [70]. The overall global prevalence of *Cryptosporidium* infection in dogs was estimated at 8% [71].

These long-established and accepted transmission routes (zoonotic and anthroponotic) of *Cryptosporidium* spp. in both different animal species and humans still require further exploration of transmission patterns and what they mean for health and public and veterinary interventions.

In our opinion, it is too common for publications to mention only the serious effects that cryptosporidiosis has on the health of children or immunocompromised people and to link these figures to the potential for zoonotic transmission of this parasite, especially in low-income or underdeveloped countries. In these situations, we need to know the causes of transmission. As we have shown in this review, even if cryptosporidiosis is generally found in crowded and unsanitary places, it is also reported in urban areas.

As we pointed out in the previous paragraphs, the prevalence of infection in Romania is high, regardless of the epidemiological sources analyzed, which we believe adds value to the specialized literature.

Moreover, we must be aware of the risk of outbreaks of cryptosporidiosis in any geographic area and the sources of infection. In addition, good collaboration between biologists, public health workers, veterinarians, and doctors is needed to be able to limit and/or prevent this infection.

Early and reliable diagnosis is necessary not only to detect infections with fatal consequences but also asymptomatic infections. In Romania, Cryptosporidium spp. infection screening is not a routine examination in clinical laboratories; therefore, most of the studies are surveys based on convenient samples collected from hospitalized children or adults. The *Cryptosporidium* spp. infection in studies analyzed in this review was diagnosed by direct microscopic stool examination [11,12] and/or modified Ziehl–Nielsen staining [11,13,14,15,16,17]. Other diagnostic methods were enzyme-linked immunosorbent assay (BIO K 070, BIO K I55—Bio-X Diagnostics, Jemelle, Belgium) [11,12], immunochromatography qualitative assay (RIDA Quick Cryptosporidium/Giardia Combi) [18], and immunofluorescence assay (Meridian) (Table 1), but none of these studies compared diagnostic techniques.

The scientific literature [20,72,73,74,75] addresses data on the different methods that are easily used for the quantification and detection of *Cryptosporidium* spp., the disadvantages of these methods, and the necessary improvements.

Omoruyi et al. compared three diagnostic techniques for Cryptosporidium infection: modified Ziehl–Nilsen staining, sandwich antigen detection enzyme-linked immunosorbent assay (sad-ELISA), and polymerase chain reaction (PCR) techniques to observe their sensitivity, specificity, and predictive values. The study included 35 HIV-positive patients with diarrhea, 125 HIV-negative patients with diarrhea, and 20 apparently healthy controls. They observed that the least sensitive technique in the studied population was the ZN staining technique; however, they emphasized the advantage of being the only technique that indicates active infections, different from ELISA and PCR techniques that cannot distinguish between active infections and inactive ones. Analyzing the costs and the necessary experience of the staff, the coloring technique with ZN also requires less expertise and costs less to apply, although using this technique means that more cases of cryptosporidiosis will go undiagnosed because it is less sensitive. Moreover, a combination of the ZN staining technique with either the sad-ELISA or PCR techniques would be the “gold standard” because the specificity and sensitivity would be very high, and thus Cryptosporidium infections would not remain undiagnosed. Finally, the incidence of Cryptosporidium infection was noticeably higher in HIV-positive individuals, hence the need to monitor Cryptosporidium oocysts in HIV-positive individuals to help provide more effective therapy [72].

The same conclusions regarding the advantages/disadvantages of diagnostic methods were also observed in a study that analyzed all epidemiological studies in Brazil, namely that the use of real-time PCR, together with microscopy and immunochromatography techniques, would result in a more precise diagnosis of cryptosporidiosis [75].

Therefore, there is a great need for rapid and easy-to-use detection methods to detect and provide information on the various *Cryptosporidium* spp. present in humans and animals. Moreover, given the fact that cryptosporidiosis is underdiagnosed and underreported, it is necessary to increase the capacity for routine surveillance and improve safety against it. This review is also intended to stimulate research that could lead to further improvements and developments in monitoring diagnostic methodologies that will aid in the more accurate diagnosis of Cryptosporidium infection.

## 4. Conclusions

Since 1983, when the first report of a human *Cryptosporidium* spp. infection in Romania was made, understanding the epidemiology has increased substantially. This review discusses many studies and shows that cryptosporidiosis is an important public health problem in our country. Although infection was first recognized as a problem in children or immunocompromised people, we observed that it can also be found in people with normal immunological function (9.1% in healthy children and 73% in immunocompromised children). The infection rates of *Cryptosporidium* spp. in immunocompromised adults from Romania are high (1.8–50%). Analyzing the prevalence of *Cryptosporidium* spp. infection in animals, we found values of 28.52% in cattle, 18% in buffalo calves, between 27.8 and 60.4% in pigs, 52.7% in dogs, and 29.4% in cats. Furthermore, in Romania, the burden of cryptosporidiosis, including acute infection and long-term sequelae, is currently unknown. The existence of strict monitoring systems is necessary, as is the education of medical personnel in the early detection of this infection and the recognition of the sources of infection, especially in vulnerable people.

## Figures and Tables

**Table 1 microorganisms-11-01793-t001:** Methods for diagnosis of *Cryptosporidium* in humans.

Test Abbreviation	Manufacturer	Method	Reference/Citation in the Present Review
ELISA IgA, IgG antibodies	In house	ELISA used to detect IgA and IgG antibodies	[21]
IFA	Meridian Laboratories, Cincinnati	immunofluorescent assay	[21]
ELISA BIO K 070	Bio-X Diagnostics, Belgium	Enzyme linked immunosorbent assay	[11,22]
ELISA BIO K I55	Bio-X Diagnostics, Belgium	Enzyme linked immunosorbent assay	[12]
ICG		RIDA Quick Cryptosporidium /Giardia Combi-immunochromatographic qualitative assay	[18]
ELISA	Nova-Tech Germany	Enzyme linked immunosorbent assay	[23]
PCR	Qiagen, Courtaboeuf, France	polymerase chain reaction	[23]
DBS-IgG	Not stated	dried blood spots for IgG	[24]
WB	Not stated	Western blot	[24]
	Not stated	Direct microscopy, modified Ziehl-Neelsen staining method, trichrome stain	[11,12,13,15,18,21,22,23,24]
PCR-RFLP analyses of the small subunit rRNA gene (18S)		Polymerase chain reaction-restriction fragment length polymorphism	[25]

**Table 2 microorganisms-11-01793-t002:** *Cryptosporidium* prevalence in Romanian population.

Year	Population	Area	No. Tested	Test	No. Positive (%)	Reference
1980–1982	Children		68		12.3% children with diareea; 9.1% of healthy children	[29]
	children and adults—presenting acute or prolonged gastroenteritis and also from asymptomatic ones, but with professional, immunological or pharmacological risk for infection		481	modified Ziehl-Neelsen staining method and Heine stainings	12 cases (2.48%)	[13]
1990–1997	HIV-POSITIVE Children who died	Bucharest—Carol Davila University of Medicine and Pharmacy	167		1.87%	[30]
1991	children (12–52 months) institutionalized at Colentina Hospital (Bucharest, Romania) and at the Dystrophic Center Vidra	Bucharest, Vidra	92	immunofluorescent assay trichrome staining	Cryptosporidium oocysts were detected in 11 fecal specimens(18%) from children at Colentina Hospital compared with no specimens from children at the Dystrophic Center	[21]
1991	Children (12–52 months) institutionalized at Colentina Hospital (Bucharest, Romania) and at the Dystrophic Center Vidra	Bucharest, Vidra	92	ELISA IgA IgG	At Colentina Hospital 42 children (70%) had IgA antibodies toCryptosporidium, 35 (58%) had IgG antibodies to Cryptosporidium, and 44 (73%) had IgA or IgG antibodies to Cryptosporidium. At the Dystrophic Center 20 children (63%) had IgA antibodies to Cryptosporidium, 16 (50%) had IgG antibodies to Cryptosporidium, and 23 (72%) had IgA or IgG antibodies	[21]
	hospitalized children with acute diarrheal disease		123		4 (3.2%)	[26]
	Children (3–6 years) physical and psychic handicaps		231		7(3.03%)—incidence	[27]
2000–2005	Children + Adults (1–64 years)	Bucharest			0.36–1.81% Incidence.	[31]
2007–2008	Children + Adults	Arad, Caras Timis counties	421	ELISA	4.03%	[22]
2008	Children + Adults 51 dystrophic infants; 49 children institutionalized within the Neuropsychiatric Recovery and Rehabilitation Timişoara; 51 children belonging to the general population, with the presumptive diagnosis of Giardiasis; 70 patients hospitalized in the Infectious and Complexes of Lugoj and Pneumo-Tuberculous Disease Hospital in Timişoara	Lugoj, Timisoara	221	Direct microscopy modified Ziehl-Neelsen staining method, ELISA	2.26%	[11]
2007–2009	patients had been hospitalized in different clinics of the Emergency Clinical County Hospital in Cluj-Napoca, Romania	Cluj Napoca	960	optical microscopy RIDA Quick Cryptosporidium/Giardia Combi-immunochromatographic qualitative assay	0	[18]
3 years	Children + adults hospitalized patients with diarrhoea	Bucharest	756	saline/iodine examination, ELISA PCR	30 cases 3.97%	[23]
2009	Children + Adults 1–80 years Institution and the Children Hospital “Louis Turcanu”, of Timisoara, Timis County, Romania and from pacients diagnosed with infectious diseases characterized by imunodepresion (HIV, lung TBC) treated in the City Hospital of Arad, Arad County, Romania.	Timis county, Arad county;	54	direct smear method modified Ziehl-Neelsen staining method	16.66%	[15]
	Children and adults with diarrhea attending to different hospitals located in Banat region	Banat region	78 (52 children and 26 adults)	modified Ziehl-Neelsen staining method PCR-RFLP	5 cases 6.41%	[22]
	Children 1 month- 14 years Children Hospital “Louis łurcanu”, of Timisoara, Timiscounty, Roumania.	Timis county	164	ELISA	4.26%	[28]
2014	Children (0–14 years) taken from children in hospitals, clinics and foster institutions.	Timis county	212	Direct fecal smears methodELISA	16 cases (7.54%)	[12]
	Children (4–13 years)	Bucharest	166	direct microscopydried blood spots for IgGWestern blot	0 in stool exam, 4 cases serology (2.41%)	[24]
2017–2020		Municipality of Iasi	11,001	modified Ziehl-Neelsen staining method	2.8%	[16]

**Table 3 microorganisms-11-01793-t003:** Methods for the detection of *Cryptosporidium* in animals.

Test Abbreviation	Manufacturer	Method	Reference/Citation in the Present Review
	Not stated	HEA, PAS, Giemsa, toluidine blue stains	[8,36]
	Not stated	Giemsa, Heine and the modified Ziehl-Neelsen staining methods	[7]
	Not stated	Flotation in a saturated solution of sodium chloride	[29]
ELISA BIO K 070	Bio-X Diagnostics, Belgium	Enzyme linked immunosorbent assay	[15,22]
	Not stated	Immunohistochemical and immunofluorescence detection	[15]
	Not stated	Direct microscopy, Ziehl-Neelsen staining methods modified by Henriksen	[8,35,37]
PCR-RFLP analyses of the small subunit rRNA gene (18S)	Not stated	Polymerase chain reaction-restriction fragment length polymorphism	[17,25]
	Philips 301 Microcroscope	Electron microscopy	[8]

**Table 4 microorganisms-11-01793-t004:** Surveys of *Cryptosporidium* infection in calves.

Study Year	Screened Region of the Country	Monitored Livestock	No. of Tested	Screening Method	No. of Positive (%) Samples	Available Epidemiological Data	References
1983	South	calves	No data	Giemsa, Heine, Ziehl-Neelsen, modif by Henriksen	No data (33–100)	a, f	[7]
1980–1982	South	calves	443	Olteanu	298 (67.3)	f	[29]
1988–1995	Western	calves	4277	Ziehl-Neelsen, modif by Henriksen	1642 (38.4)	a, f	[8]
2005–2009	Western	calves	420	ELISA	175 (41.6)	a, e	[37]
2007	Western	calves	60	ELISA	49 (77.7%)	a, e	[40]
2007–2008	Center, Nord-Western	calves	425	Ziehl-Neelsen, modif by Henriksen	121 (28.52)	a, e	[42]
2008–2009	Western, Center, Nord-Western	calves	370	ELISA	153 (41.4)	a, f	[43]
2011	Banat	calves	258	Ziehl-Neelsen, modif by Henriksen	65 (25.0)	a, c, e, f	[44]

a = age; c = breed; e = habitat; f = diarrhoea.

**Table 5 microorganisms-11-01793-t005:** Surveys of *Cryptosporidium* infection in sheep, goats, and pigs.

Study Year	Screened Region of the Country	Monitored Livestock	No. of Tested	Screening Method	No. of Positive (%) Samples	Available Epidemiological Data	References
1980–1982	southern	sheep	383	Olteanu	244 (63.71)	a, f	[29]
2010	western	lambs	66	ELISA	6 (9.09)	a, f	[15]
2011	western	sheep	116	Direct exam the smears of faeces	15 (12.9)	a, d, f	[49]
2013	western	sheep	175	Ziehl-Neelsen, modif by Henriksen	24 (13.7)	a, f	[47]
1980–1982	south	goats	143	Olteanu	110 (76.9)	a, f	[29]
1986–1988	center	goats	120	Willis	(10–60)	a, d, e	[46]
1980–1982	south	pigs	743	Olteanu	449 (60.4)	a, f	[29]
2008	western	pigs	97	ELISA	32 (33.0)	a, e	[50]
2010	western	pigs	133	ELISA	37 (27.8)	a	[37]

Legend: a = age; d = diet; e = habitat; f = diarrhoea; ELISA—Enzyme-Linked Immunosorbent Assay.

**Table 6 microorganisms-11-01793-t006:** Surveys of *Cryptosporidium* infection in dogs and cats.

Study Year	Screened Region of the Country	Investigated Pets	No. of Tested	Screening Method	No. of Positive (%) Samples	Available Epidemiological Data	References
2008–2009	center	dogs	374	ELISA	197 (52.7)	a, e	[52]
2008	western	dogs	63	ELISA Ziehl-Neelsen, modif by Henriksen	0 (0.0)	a, b, f	[37]
2008	center	cats	180	ELISA+ Ziehl-Neelsen	53 (29.4)	a, b, e	[53]

a = age; b = sex; e= habitat; f = diarrhoea.

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
