# Peer review of "Epidemiology of Cryptosporidium Infection in Romania: A Review"

_microorganisms, 2023, doi:10.3390/microorganisms11071793_

Round 1
Reviewer 1 Report
The authors reviewed the epidemiology of Cryptosporidium infection in Romania. This may be a relevant topic but the manuscript is lacking many explanations and comparisons. Currently it is a simple listing of results from Romania but without any perspective and comparison to general knowledge.
This started with the lack of a working hypothesis. We need to know why we should bother with cryptosporidiosis in Romania. Are there reasons to expect that it is different to neighboring countries? Are the farming conditions different? Some explanations would be highly appreciated.
The manuscript is lacking a discussion section. There are recommendations how cryptosporidiosis should be diagnosed or Cryptosporidium species identified. These recommendations have to be discussed and put in perspective to reports mentioned in the review. Cryptosporidium species have been detected all over the world in a multitude of organisms. These results have to be compared to reports from Romania. Only with a decent discussion of the Romanian reports the reader will be able to judge what is similar and what is different and why. In this context I am referring to several excellent overviews, here are a few:
Helmy, Y. A., & Hafez, H. M. (2022). Cryptosporidiosis: From Prevention to Treatment, a Narrative Review. Microorganisms, 10(12), 2456.
Pane, S., & Putignani, L. (2022). Cryptosporidium: still open scenarios. Pathogens, 11(5), 515.
Luka, G., Samiei, E., Tasnim, N., Dalili, A., Najjaran, H., & Hoorfar, M. (2022). Comprehensive review of conventional and state-of-the-art detection methods of Cryptosporidium. Journal of Hazardous Materials, 421, 126714.
Chen, Y., Qin, H., Huang, J., Li, J., & Zhang, L. (2022). The global prevalence of Cryptosporidium in sheep: a systematic review and meta-analysis. Parasitology, 1-44.
Hijjawi, N., Zahedi, A., Al-Falah, M., & Ryan, U. (2022). A review of the molecular epidemiology of Cryptosporidium spp. and Giardia duodenalis in the Middle East and North Africa (MENA) region. Infection, Genetics and Evolution, 105212.
Zuo, M. R., Li, X. T., Xu, R. Z., Sun, W. C., Elsheikha, H. M., & Cong, W. (2023). Global prevalence and factors affecting the level of Cryptosporidium contamination in soil: A systematic review, meta-analysis, and meta-regression. Science of The Total Environment, 164286.
Chen, Y., Wu, Y., Qin, H., Xu, H., & Zhang, L. (2023). Prevalence of Cryptosporidium infection in children from China: a systematic review and meta-analysis. Acta Tropica, 106958.
Li, X. M., Geng, H. L., Wei, Y. J., Yan, W. L., Liu, J., Wei, X. Y., ... & Liu, G. (2022). Global prevalence and risk factors of Cryptosporidium infection in Equus: A systematic review and meta-analysis. Frontiers in Cellular and Infection Microbiology, 12, 1774.
Yin, J., Shen, Y., & Cao, J. (2022). Burden of Cryptosporidium Infections in the Yangtze River Delta in China in the 21st Century: A One Health Perspective. Zoonoses.
Omolabi, K. F., Odeniran, P. O., & Soliman, M. E. (2022). A meta-analysis of Cryptosporidium species in humans from southern Africa (2000–2020). Journal of Parasitic Diseases, 46(1), 304-316.
Furthermore:
Abstract: The conclusion in very vague and in this form almost meaningless. Please sharpen your statement.
Always set the names of genera and species in italic.
Line: next 50 years? Better write last 50 years
Lines 123: this sentence is incomplete
Lines 259-261: Please delete 5. Patents because here nothing has been reported.
some minor mistakes
Author Response
We sincerely thank the Editor and reviewers for their thoughtful recommendations for improving the quality of our manuscript ("Epidemiology of Cryptosporidium infection in Romania: a review"). We have revised the manuscript based on the comments and suggestions of the reviewers. We have also provided point-by-point responses to the reviewers’ comments, and the changes made according to the reviewer’s suggestions are shown in red highlight in the enclosed revised manuscript
Please see the attachment

Reviewer 2 Report
The manuscript summarized the published data on Cryptosporidium detection methods and infection rates in humans and animals in Romania. It is very useful for scientists to understand the situation in Eastern European countries.
Major comments:
1. Abstract : The major results in this study, such as infection rates of Cryptosporidium in different hosts should be provided. The same is true for conclusion.
2. Conclusion: the last two sentences should be deleted as they referred to significance of this study.
3. Introduction: The info for the third paragraph is not exactly true and this paragraph is suggested to be deleted as it is not related to the results section.
4. Results: the reason for the broad range of infection rates within different hosts should be analyzed.
5. Table 2: In the column named "paragraph", detailed description should be replaced with short description, such as healthy children, children with diarreah etc.
Minor comments:
1. Cryptosporidium and the species within this genus should be italized throughout the report
2. Line 123: "was" at the end this line should be deleted
3. Lines 166-167: should be combined into the same line
the writing is fine
Author Response
We sincerely thank the Editor and reviewers for their thoughtful recommendations for improving the quality of our manuscript ("Epidemiology of Cryptosporidium infection in Romania: a review"). We have revised the manuscript based on the comments and suggestions of the reviewers. We have also provided point-by-point responses to the reviewers’ comments, and the changes made according to the reviewer’s suggestions are shown in red highlight in the enclosed revised manuscript
"Please see the attachment."

Round 2
Reviewer 1 Report
I can see that the authors did substantial improvements to their article, however, there are still some concerns. In order to understand the problem of cryptosporidiosis in Romania, the authors have to discuss the bias caused by different detection methods. If there are false negative or false positive detections made by impropriate methodology the entire statistics becomes distorted. What can be learned here from other countries/studies? Please elaborate.
Lines 69ff: it is not correct that the detection methods must be individualized from country to country. There should be one optimal detection method for all, see e.g. Luka, G., Samiei, E., Tasnim, N., Dalili, A., Najjaran, H., & Hoorfar, M. (2022). Comprehensive review of conventional and state-of-the-art detection methods of Cryptosporidium. Journal of Hazardous Materials, 421, 126714.
Line 96: replace move by isolate
Always set the names of genera and species in italic.
Always set the reference before the full stop (e.g. [23]. Instead of .[23]).
There are some language problems and corrections should be made by a native speaker.
There are some language problems and corrections should be made by a native speaker.
Author Response
We did all the suggestions you mentioned in our new manuscript.
We addressed, as you suggested in the Discussions section, regarding the data on the different methods that are used for the quantification and detection of Cryptosporidium spp. both in Romanian and worldwide studies.
Thank you for your comments and we hope to meet your expectations. We have done a thorough English grammar and spelling check and provided the necessary changes.
